# Testing the validity and adequacy of linguistic phylogenetic analyses

Benedict King ⃝ *

Department of Linguistic and Cultural Evolution, Max Planck Institute for Evolutionary Anthropology, Leipzig, Saxony, Germany

* benedict_king@eva.mpg.de

## Abstract

Bayesian phylogenetics has become a standard tool in historical linguistics, and for the most part implements models borrowed from evolutionary biology. Not enough work has been done to validate the analysis set-up that has become standardised in phylolinguistics, which consists of binary data with ascertainment bias, data partitions with correlated cognate count and rate, the binary covarion substitution model, and the uncorrelated lognormal branch rate model. Here I perform a set of simulation-based calibration studies to test a typical phylolinguistic analysis in the software BEAST2. Although the analysis can correctly recover the parameters of the substitution model, complications arise due to the combination of ascertainment bias and partitions of unequal length and rate. Reweighting the partition-specific rates by the number of cognates, as is the default behaviour, leads to poorly calibrated posteriors. An alternative approach, where each meaning is assumed to come from a set of cognates of equal size, behaves correctly in simulations and is found to fit better to empirical data. I also assess the adequacy of the covarion substitution model through posterior predictive simulations. The covarion is found to fall short of approximating the true process of lexical evolution, likely due to the prevalence of semantic shift and the non-independence of cognate substitutions in real data. This work highlights the importance of thorough testing of models and their implementation in phylolinguistics, as well as the need for further research on improving models of lexical evolution.

## Author summary

Linguists use Bayesian inference to construct language trees. Given the prevalence of these methods, it is vital that they are tested, so that we can have confidence in the results. Bayesian inference should produce results that are "well-calibrated", for example predictions with an 80% probability should be true 80% of the time. A correctly implemented analysis will be well-calibrated by design, with any deviation from good calibration indicative of problems in the

**Data availability statement:** All data and code supporting this study are available at https://doi.org/10.6084/m9.figshare.30870404.

**Funding:** BK is supported by the Max Planck Institute for Evolutionary Anthropology (https://www.eva.mpg.de). The funders had no role in study design, data collection and analysis, decision to publish, or preparation of the manuscript.

**Competing interests:** The author has declared that no competing interests exist.

analysis set-up or software bugs. Here I perform a set of calibration experiments for typical phylolinguistic analyses. The analyses are well-calibrated, but only after a change to the default model set-up for handling meaning-specific rates. I also test the adequacy of linguistic phylogenetic models, which refers to how accurately the models reflect real-world processes. In several quantifiable ways, data simulated under the model differs significantly from the real data. This is likely driven by the non-independent evolution and widespread semantic shift that characterises real lexical data. This work highlights that although the implementation of phylolinguistic models is valid, in that the correct parameter estimates are returned for a given model and dataset, more work is needed to test if the models themselves are appropriate.

## Introduction

Bayesian phylogenetic analysis of linguistic data is prevalent in the historical linguistics literature [1–30]. A set of standard models, implemented in the software package BEAST2 [31], has emerged for phylogenetic analysis of lexical data [32]. The data for a phylolinguistic analysis is usually cognate-coded basic vocabulary data [33]. Lexemes, specifically the canonical word form, are collected for each language in a given set of meanings. A list of the most stable meanings is chosen (a Swadesh list or variations thereof), which typically has around 150–200 items. These lexemes are then cognate-coded, so that each item is assigned to a class based on common descent.

Two main possibilities exist for how the cognate coded lexical data is structured for analysis: multistate and binarised. In a multistate analysis a dataset of $N$ meanings is represented as $N$ phylogenetic characters with $k_i$ states, where $k_i$ represents the number of cognate classes in the meaning $i$. This multi-state approach has only occasionally been applied to linguistic data [3,19,34]. Instead, the standard approach in phylolinguistics is binarised data (Table 1). Each cognate is treated as a character with two states: absent (0) and present (1), and treated independently from other cognates in the same meaning. There will then be $\sum_i^N k_i$ characters. The data consists of $N$ meaning partitions, within which there will be a varying number of cognate sets. The number of cognate sets in a meaning, and therefore the length of the corresponding data partition, is dependent on the stability of the meaning. The faster the rate of lexical replacement, the more cognate sets. This can be observed in Table 1, where the stability of the numeral *two* means all the lexemes are cognate, whereas lexemes for the meaning *belly* fall into four different cognate sets.

It follows from the way the data is collected that there are no cognate sets absent for every language. This is known as ascertainment bias, and must be corrected for during the likelihood calculation [35]. The ascertainment bias correction involves a re-normalisation of the likelihood function to account for the fact that all-absent cognates are unobservable. For a given pattern $x$ (a pattern refers to a unique configuration of absent and present states across languages) the likelihood is normalised thus:

**Table 1. Example of binarised cognate data for two meanings and five Slavic languages.**

| concept | two | belly | | | |
|---|---|---|---|---|---|
| cognate | *dṷo- | *terh₁- | *bʰreṷs- | *gʷi̯eh₃- | *kʷeh₂l- |
| Slovene | 1 | 1 | 0 | 0 | 0 |
| Polish | 1 | 0 | 1 | 0 | 0 |
| Czech | 1 | 0 | 1 | 0 | 0 |
| Ukrainian | 1 | 0 | 0 | 1 | 0 |
| Rusyn | 1 | 0 | 0 | 0 | 1 |

$$Pr(x \mid observable) = \frac{Pr(x)}{1 - Pr(unobservable)}$$

From this equation it is apparent that when the likelihood of an all-absent cognate is high (for example due to a low evolutionary rate), the renormalisation procedure increases the likelihood of the observed patterns due to the smaller denominator. Failure to apply this correction leads to over-estimation of the substitution rate [36]. In contrast to morphological phylogenetics, where the correction is for invariant characters [37], for cognate data ascertainment bias correction is only for cognates that are *absent* in all languages. Cognates that are present in all languages, although invariant, can and are observed: because phylolinguistics draws upon a fixed set of meanings, occasionally all lexemes in a particular meaning are cognate. Note that for empirical datasets with missing data, the ascertainment bias correction is slightly modified, since the probability of an all-absent increases as the proportion of missing data increases [7].

A handful of different models have been applied to the analysis of binarised lexical data, namely the binary [1,35], covarion [38,39] and Pseudo-Dollo [40] models. Where multiple models have been compared, the binary covarion has usually been the best-fitting model [3,4,10,11,14,18,20,24,28,30], but see [8]. As it is also by far the most commonly used model for linguistic data, it is the focus of the present study.

The binary covarion model divides absent and present states into additional fast and slow states, making a total of four states: absent-fast, present-fast, absent-slow and present-slow. The parameters of the covarion model are $\alpha$, which determines the rate of the slow state relative to the fast state, $s$, the switching rate between fast and slow states and the equilibrium frequencies $\pi_{obs}$ and $\pi_h$. Equilibrium frequencies determine the relative frequencies of the states after the process runs for an infinite length of time. The frequencies $\pi_{obs}$ of the observed states and the hidden states ($\pi_h$) are combined to provide the equilibrium frequencies of the four states. The full unnormalised matrix of instantaneous transition rates (from row to column) between the states is then given by:

$$Q = \begin{pmatrix} - & \pi_{obs}[1]\pi_h[0] & s\pi_{obs}[0]\pi_h[1] & 0 \\ \pi_{obs}[0]\pi_h[0] & - & 0 & s\pi_{obs}[1]\pi_h[1] \\ s\pi_{obs}[0]\pi_h[0] & 0 & - & \alpha\pi_{obs}[1]\pi_h[1] \\ 0 & s\pi_{obs}[0]\pi_h[0] & \alpha\pi_{obs}[0]\pi_h[1] & - \end{pmatrix}$$

The implementation of the binary covarion model in BEAST2 has three different modes. The Q matrix above demonstrates the fully time-reversible set-up, which includes the frequencies of the hidden states within the Q matrix. In the default "beast" mode, the hidden frequencies are excluded from the Q matrix, which is then only time-reversible if the fast and slow hidden states have equal frequencies (0.5, 0.5). Finally, in the original "Tuffley-Steel" mode, the parameterisation includes separate switch rates for fast and slow modes, and the $\alpha$ parameter is fixed at a value of 0. For this study, I used the default set-up, prevalent within phylolinguistics, which is the standard "beast" parameterisation with hidden frequencies set at (0.5, 0.5).

PLOS Computational
Biology

As mentioned above, the number of cognate sets in a meaning partition is dependent on the rate of lexical replace-ment. Higher rates means more cognate sets. One way of modelling this is to apply different rate multipliers to each meaning partition. These rates are hereafter referred to as "partition rates", but are often called "mutation rates" within the BEAST2 software. The partition rates are drawn from a Dirichlet distribution, which, if unweighted, has a mean of 1 across the rates. However, partitions have different numbers of cognates, necessitating a reweighting procedure. Reweighting ensures that the overall mean rate across observed cognates is 1, taking into account that each partition rate is applied to a different number of cognates. The default reweighting procedure, implemented in BEAST2 analyses generated by the BEAUti software, sets the weights of the delta exchange operator equal to the number of cognates in each partition. However, the cognate sets are an ascertained sample, with all-absent cognates effectively filtered out. Because of this, the separate operations of ascertainment bias correction and partition rate reweighting may interact, the consequences of which are unexplored.

Three alternative configurations of partition rates were compared in [3]: all meanings given the same rate (no additional free parameters), an individual rate applied to every meaning partition (169 free parameters), and 8 different rates applied to meanings in bins according to the number of cognates (7 additional free parameters). The latter, binned model was found to be the best fitting.

In summary, Bayesian phylogenetic analysis of linguistic data combines the following features:

- Binary presence/absence data

- The binary covarion substitution model

- Ascertainment bias (all-absent cognates are not observed)

- Meaning partitions consisting of variable numbers of cognates

- Different rates applied to meaning partitions

- Correlation of partition length and partition rate

Recently there has been an increased focus on improved testing and validating of evolutionary biology software [41], including Bayesian phylogenetic models [42,43]. Because linguistic phylogenetic models borrow methodology from vari-ous sources within evolutionary biology, the particular combination of model aspects detailed above have not been thor-oughly tested as a combination. Indeed, the *babel* package in BEAST2, which contains a number of features for analysing linguistic data specifically, has no associated publication.

Simulation-based calibration [44] is emphasised as a central method for validating phylogenetic models [43]. An imple-mentation of a Bayesian model is well-calibrated when predictions with probability X% are correct X% of the time [45]. The procedure involves sampling parameters from their prior probability distributions, simulating data from these parameters, and analysing the simulated datasets under the same model. The results are then checked for coverage: for example the correct value for a parameter should lie within the 95% highest posterior density interval 95% of the time.

A second test is Rank Uniformity Validation (RUV). It has been shown that any particular draw of parameter values from the prior distribution is also a draw from the posterior distribution conditional on data generated by that same prior draw [46]. In other words, in SBC, the true value used to generate the data (the prior draw), and the parameter estimates pro-duced when that data is analysed (the corresponding posterior draws), should come from the same distribution provided the analysis is correctly implemented. It follows that when the posterior draws are ordered by rank, the true value has an equal probability of falling between any two of the posterior draws. The rank of the true value within the corresponding pos-terior draws should follow a uniform distribution (S1 Fig). Rank Uniformity can also be visualised using an empirical cumu-lative distribution (ECDF) plot [47]. The rank of the true values within their respective posterior distributions, normalised to a 0–1 scale (the Probability Integral Transform, or PIT), is on the x-axis. The y axis is the ECDF, and shows cumulative

proportion of PIT values equal to or less the value on the x axis. Under rank uniformity, the ECDF plot should follow the diagonal. An ECDF difference plot, depicting departures from the expected value, more clearly shows any violations of rank uniformity. Examples of ECDF difference plots showing various kinds of departure from RUV are shown in S2 Fig.

Beyond validating the implementation of phylolinguistic analyses, there is also the question of model adequacy. Whereas validation ensures correctness of the computation in the inference machinery, adequacy refers to the how well the model represents the true process. It is possible that an analysis can be valid, in that the correct parameter estimates for a given model and dataset are estimated, but nevertheless inadequate if the model is mismatched to the underlying process. In phylolinguistics, model adequacy is a question of how accurately the covarion model approximates the true process of lexical evolution.

The absolute adequacy of phylogenetic models can be assessed using posterior predictive simulations [48–50]. In posterior predictive simulation, simulations are performed drawing on parameters from the posterior distribution, i.e., those estimated from the empirical data. If the model adequately describes the true process, these simulated datasets should "look like" the empirical data, and this is quantified using one or more metrics. A large discrepancy between values for a metric calculated from the posterior predictive datasets compared to the empirical data indicates model inadequacy.

Here I assess the validity of the standard approach to Bayesian inference of lexical data using simulation-based calibration. I also perform posterior predictive simulations to assess the adequacy of the covarion model to describe the process of lexical evolution.

## Materials and methods

### Simulation-based calibration

I set up a typical phylogenetic model for linguistic data (Fig 1). The model used the binary covarion substitution model, in which the parameters are the relative rate of the slow state ($\alpha_{bcov}$), the switch rate between fast and slow states ($s_{bcov}$), the stable frequencies of the present and absent states ($\pi_{obs}$), and the stable frequencies of the hidden (fast and slow) states ($\pi_{hidden}$). To represent different data partitions with different stabilities, I drew three rates ($m_i$) from a Dirichlet distribution and simulated 3 partitions ($D_i$) of 10000 cognates using these rates. The large number of simulated cognates was necessary due to the large proportion of all-absent cognates in the simulated data. The tree of 30 tips was drawn from a Yule model with birth rate $\lambda$, and the branch rates from lognormal distribution [51]. The clock rate ($\mu$) was fixed to 0.05, so that the root age and tree length were identifiable.

To perform simulation-based calibration (SBC) I generated 100 simulated datasets using the software LinguaPhylo [42]. Simulated data differs from empirical data because all-absent cognates are present. All-absent cognates in the simulations primarily result when simulations begin in the absent state at the root, and no transitions to the present state occur along the tree. When the transition rate from absent to present is low (either due to a slow clock rate or a low stable frequency of the present state), a correspondingly larger proportion of the 10,000 simulated cognate sets are all-absent. I re-analysed simulated datasets, using the same model and priors used to generate the data, in BEAST2.7.8 [31]. To confirm that the MCMC chains had run sufficiently long, I calculated effective sample size (ESS) scores for each free parameter in each simulation replicate using the R package sns [52]. ESS scores were over 200 for all parameters in all simulation replicates.

Three versions of the SBC were performed to test the ascertainment bias correction, based on different treatments of all-absent cognate classes and the weighting of the 3 partition rates.

1. The simulated data remained unaltered, all-absent cognates retained.

2. All-absent cognates removed and ascertainment bias correction implemented. Partition rates not reweighted.

3. All-absent cognates removed and ascertainment bias correction implemented. Partition rates re-weighted (the default option in BEAST2). I applied the same weighting to the Dirichlet distribution prior from which the partition rates are drawn, and the partition rate operator during reanalysis of simulated data.

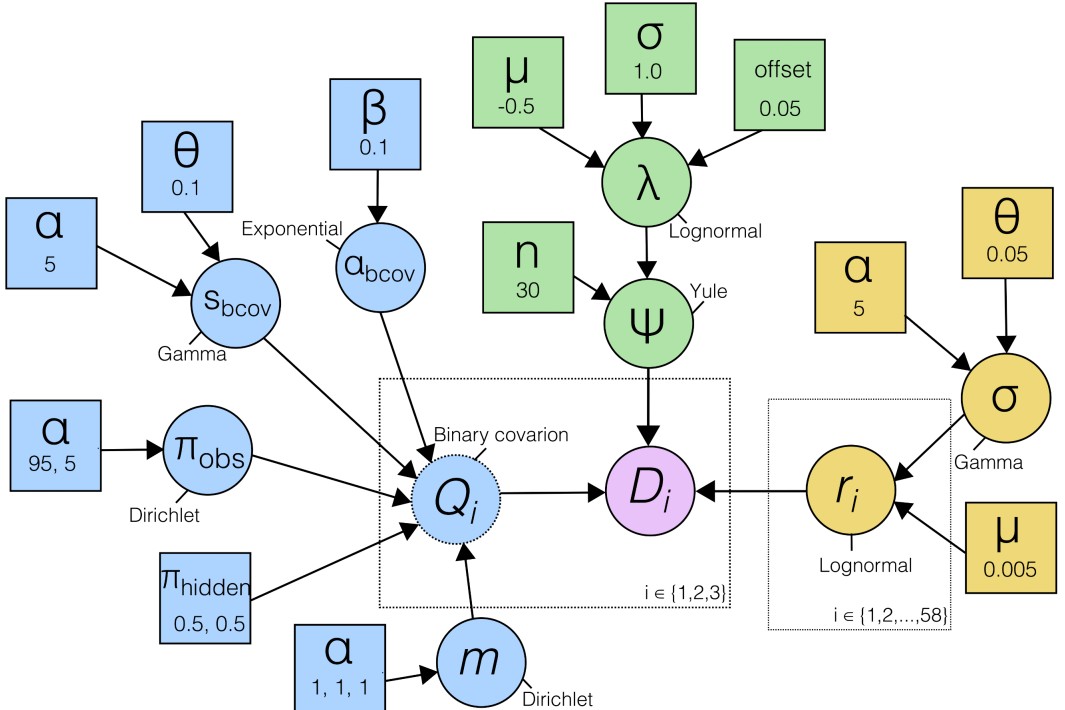

**Fig 1. Graphical model of the analysis used for Simulation-Based Calibration.** In blue, the parameters of the covarion substitution model, in green the tree model and in yellow the relaxed clock model. $i \in \{1, 2, 3\}$ and $i \in \{1, 2, ..., 58\}$ refer to iteration over partition rates and branches, respectively.

The outcomes of the Bayesian analyses were assessed for both coverage and Rank Uniformity Validation, following best practice guidelines [43]. To assess if clade posterior probabilities were well-calibrated, I first generated for each simulation replicate a list of all clades found in the posterior, then calculated their posterior probability and whether or not they were present in the true tree. Results for all 100 simulation replicates were then pooled and calibration assessed using stable reliability diagrams [53].

## Model-testing on empirical data

I tested three different model setups on a dataset of Indo-Iranic languages taken from the IE-CoR dataset [54]. The analysis set-up followed that of [3]. The 170 meaning classes were divided into five partitions based on the number of cognate sets, with bins of 1–5, 6–10, etc. Each of these partitions had a separate partition rate parameter (sometimes referred to as the mutation rate). These parameters were drawn from a weighted Dirichlet distribution and a weighted delta exchange operator was applied to propose new values.

  Three different weighting strategies were compared.

1. Weighting by the number of cognate sets in each partition. This is the default mode in BEAST2.

2. Unweighted: i.e., all partitions have equal weights

3. Weighting by the number of meanings in each partition

I calculated the marginal likelihood of these three models using path sampling [55], with 30 power posteriors estimated using parallel computation [56].

## Posterior predictive simulations

I used a sample from the IE-CoR dataset of the extant languages of the Germanic, Italic and Celtic subgroups (35 languages, 1251 cognate sets). This sample contains no missing data, therefore facilitating the comparison of the empirical and simulated datasets. I extracted the subtree of these languages from the summary tree in [3]. This tree was fixed during the analysis, and the values of the binary covarion and relaxed clock parameters were estimated in BEAST2.7.8.

I simulated 1001 posterior predictive datasets, each corresponding to a set of parameter estimates from the log file of the BEAST2 analysis. For comparison of simulated and empirical datasets I calculated 8 metrics, designed to capture key aspects of language divergence, the composition of languages (in terms of cognates), and the distribution of cognates across languages. For languages I calculated the maximum, minimum and mean pairwise distance between languages, mean proportion of present cognates and variance in the number of present cognates. For cognates I calculated the number of prevalent cognates (prevalent defined as greater than 80% of the language sample), number of singletons (cognates present in one language, autapomorphies) and variance in cognate prevalence. For each metric I calculated the mid-point two-tailed p-value [57], denoted as $p_B$. A value of 1 would indicate perfect model fit: the metric for the empirical data falls in the centre of the distribution over the posterior predictive sample. A value of <0.05 indicates a significant difference between the empirical and posterior predictive datasets.

## Results and discussion

### The binary covarion model performs well, but ascertainment bias with multiple partitions remains a difficult problem

Validation simulation study 1, in which the simulated data were not modified to remove all-absent cognates, performed well on both coverage and RUV (S3 and S4 Figs). In simulation study 2, ascertainment columns were removed resulting in partitions of varying length, but the Dirichlet prior and delta exchange operator were not reweighted during the re-analysis of simulated data. This analysis also performed correctly. Parameters of the binary covarion substitution model, the partition-specific rates, the branch rate parameters and the tree height and length were all well-calibrated (Figs 2 and 3). Clade posterior probabilities were also well-calibrated (i.e., nodes with 60% support were in the true tree 60% of the time, etc.), as shown by the reliability diagram (Fig 4).

For simulation study 3, ascertainment columns were removed, and the Dirichlet prior and delta exchange operator on partition rates reweighted according to the number of cognates in each partition. This model fails coverage and RUV checks (Fig 5). Partition rates and the birth rate are underestimated, whereas the tree height and length are overestimated. Nevertheless, estimates for binary covarion parameters, branch rate model parameters, and clade posterior probabilities remain well-calibrated.

That analysis 3 should fail calibration checks is not surprising. Reweighting the Dirichlet distribution prior for reanalysis of simulated data means that there is a small difference between the model used to simulate the data and the model used to analyse it. More importantly, reweighting the delta exchange operator prevents the analysis finding the exactly correct values for the 3 partition rates, which sum to 1 under an unweighted rather than a weighted Dirichlet distribution.

The problem when it comes to analyses of empirical data is that the number of cognate sets before the removal of ascertainment columns is not known. Therefore an exact replica of our well-calibrated analysis 2 is not possible. More to the point, the number of cognate sets including all-absent cognates is not meaningful for empirical data, since there is no such thing as a cognate set that is absent in all languages, and no set number of these.

Nevertheless, the data is analysed *as if* all-absent cognate sets exist. One way to model this is to assume that for each meaning there is a fixed number of cognate sets, some of which are absent in all languages, and the number of observed cognate sets is a function of the rate for that meaning. On an empirical dataset of Indo-Iranic languages, this analysis

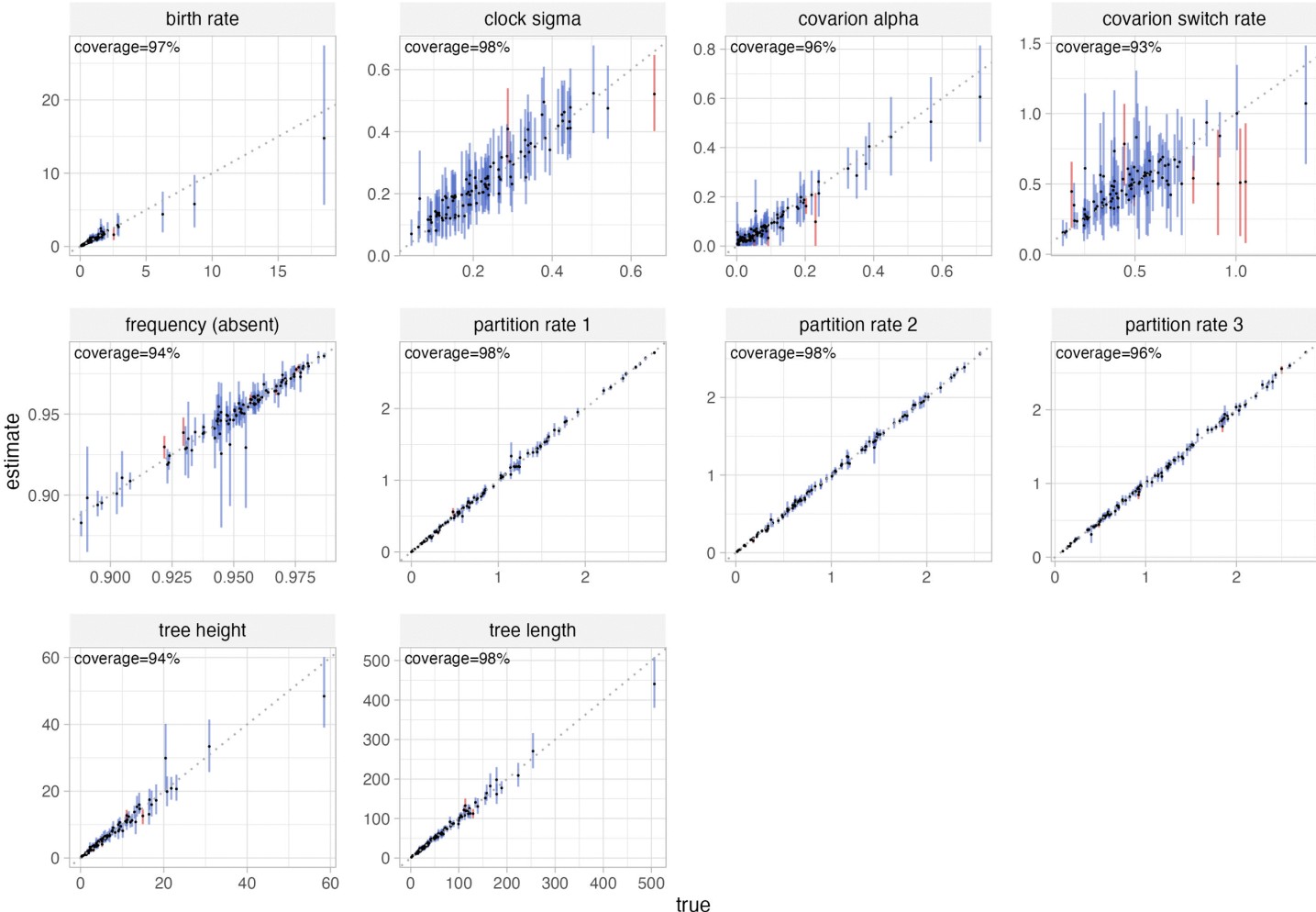

**Fig 2. Coverage plot of validation simulation study 2.** Ascertainment cognates (absent in all languages) were removed in this implementation, but partition rates were not reweighted. Each panel shows the true value from which data were simulated on the x axis, and the 95% highest posterior density (HPD) interval of the estimate of the parameter from the simulated data. The dotted diagonal lines represent x = y. Lines are blue when the true value is within the HPD, otherwise red. Coverage refers to the percentage of HPDs which contain the true value, which should be approximately 95% for a well-calibrated analysis.

set-up, in which the Dirichlet prior and delta exchange operator are reweighted by the number of meanings, outperforms both a typical set-up in which reweighting follows the number of cognates, and an unweighted set-up (Table 2).

Comparison of parameter estimates from the meaning-weighted and cognate-weighted analyses shows differences in the partition rate estimates and the clock rate, mirroring the results found in the SBC tests (S5 and S6 Figs). Most notably, the clock rate is lower in the meaning-weighted analysis. This is not surprising, since the cognate-rich meanings no longer have an outsized effect on the weighted average of the partition rates. The clock rate under a meaning-weighted model set-up requires reinterpretation. The units correspond to expected changes per cognate per unit time *in a meaning with average rate*.

The results raise questions about how previously published linguistic phylogenies, which use the default partitioning set up, have been affected. SBC tests show that problems are confined to the partition rates, birth rate, tree height and tree length (the latter three are expected to correlate). In the empirical analyses, only the clock rate differs between the

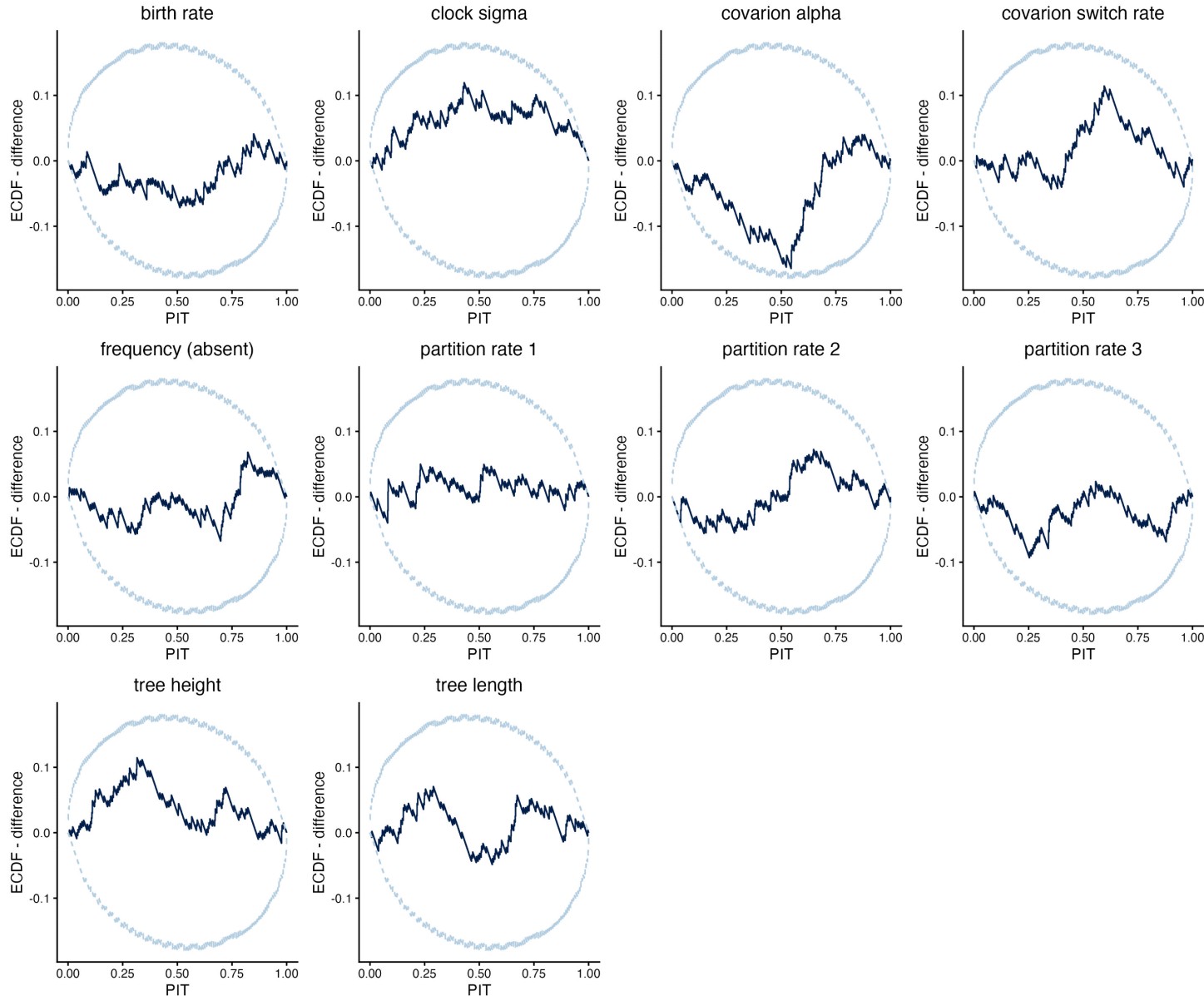

**Fig 3. ECDF difference plot of validation simulation study 2.** Ascertainment cognates were removed in this implementation, but partition rates were not reweighted. An ECDF plot shows the empirical cumulative density curve of the PIT scores for the true values within the posterior estimates from simulated data. The ECDF difference plot shows deviation from the expected ECDF curve under uniformity. Since the plot remains within the 95% confidence bands for all parameters, there is no significant deviation from uniformity, and the analysis is considered well-calibrated. Further information on interpretation is in the main text and S1 and S2 Figs.

different partitioning set-ups, whereas birth rate, tree height and tree length remain unchanged. In the SBC tests the clock rate is fixed so problems will manifest as increased tree height. In the empirical data, where clock rate and tree height are estimated, only the clock rate and partition rates differ between alternative partitioning set-ups. Since the clock rate and partition rates are rarely the parameters of interest on which the success or failure of a particular hypothesis rests, the results here do not call into question the main conclusions of previous phylolinguistic studies. Since the partition rates in

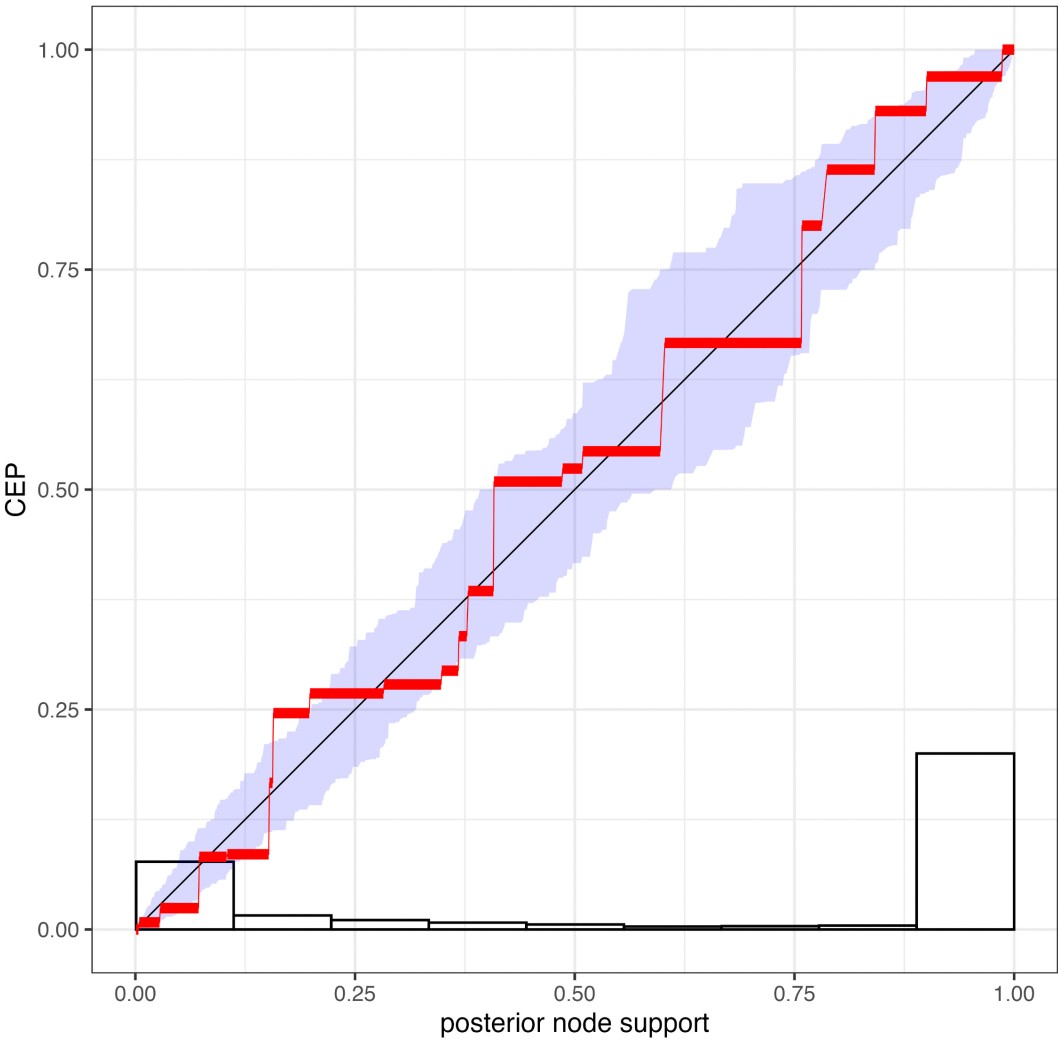

**Fig 4. Stable reliability diagram showing that node support values are well-calibrated in validation simulation study 2.** The x-axis shows the posterior node probability, and the y-axis shows the conditional event probability (CEP), the probability that a node is observed in the true tree given it's posterior support value. The stepped nature of the graph is because of binning of nodes with a range of posterior probabilities. The purple area represents 90% confidence bands. The histogram shows the frequency of nodes with different support values. Essentially, this diagram shows that nodes with a posterior probability of X% are true X% of the time.

the SBC test are drawn from distributions similar to those found in empirical analyses, problems should not necessarily increase on larger datasets than those tested.

### The binary covarion model does not fully describe all aspects of lexical evolution

Of the eight metrics used to compare the posterior predictive and empirical datasets, four show a significant difference ($p_B < 0.05$), suggesting model misspecification (Fig 6). These differences can be attributed to two main causes, violation of the assumption of stability, and non-independent evolution of cognate sets.

Violation of the assumption of stability is evidenced in the lower proportion of present cognates in the simulated data than the empirical data. Arguably, the stable frequency of the present state for any given cognate is even more extreme

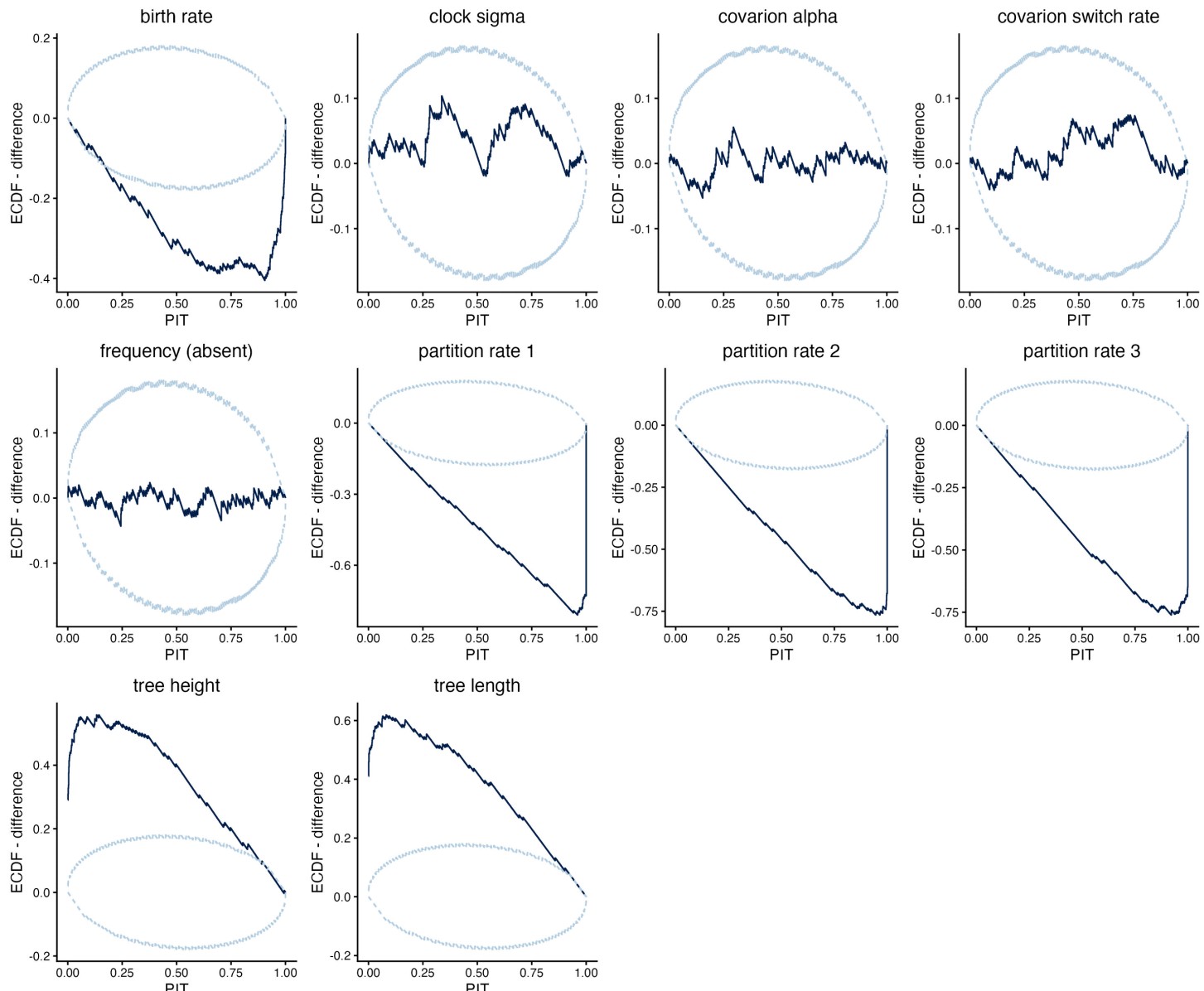

**Fig 5. ECDF difference plot of validation simulation study 3.** Ascertainment cognates were removed in this implementation, and partition rates were reweighted. The plots reveal systematic underestimation of birth rate and partition rates, and overestimation of tree height and tree length.

**Table 2. Bayes factor comparison of three different ways of weighting the parition-specific rate multipliers (mutation rates).** Bayes factors compare each model to the best-supported model. Weighting by the number of meaning classes in the partition is the best performing model, whereas weighting by the number of cognate sets performs the worst.

| model | log marginal likelihood | Bayes Factor vs best |
|---|---|---|
| reweighted by number of cognates | -15401.4 | 0.0116 |
| reweighted by number of meanings | -15396.94 | 1 |
| equal weighting | -15399.4 | 0.089 |

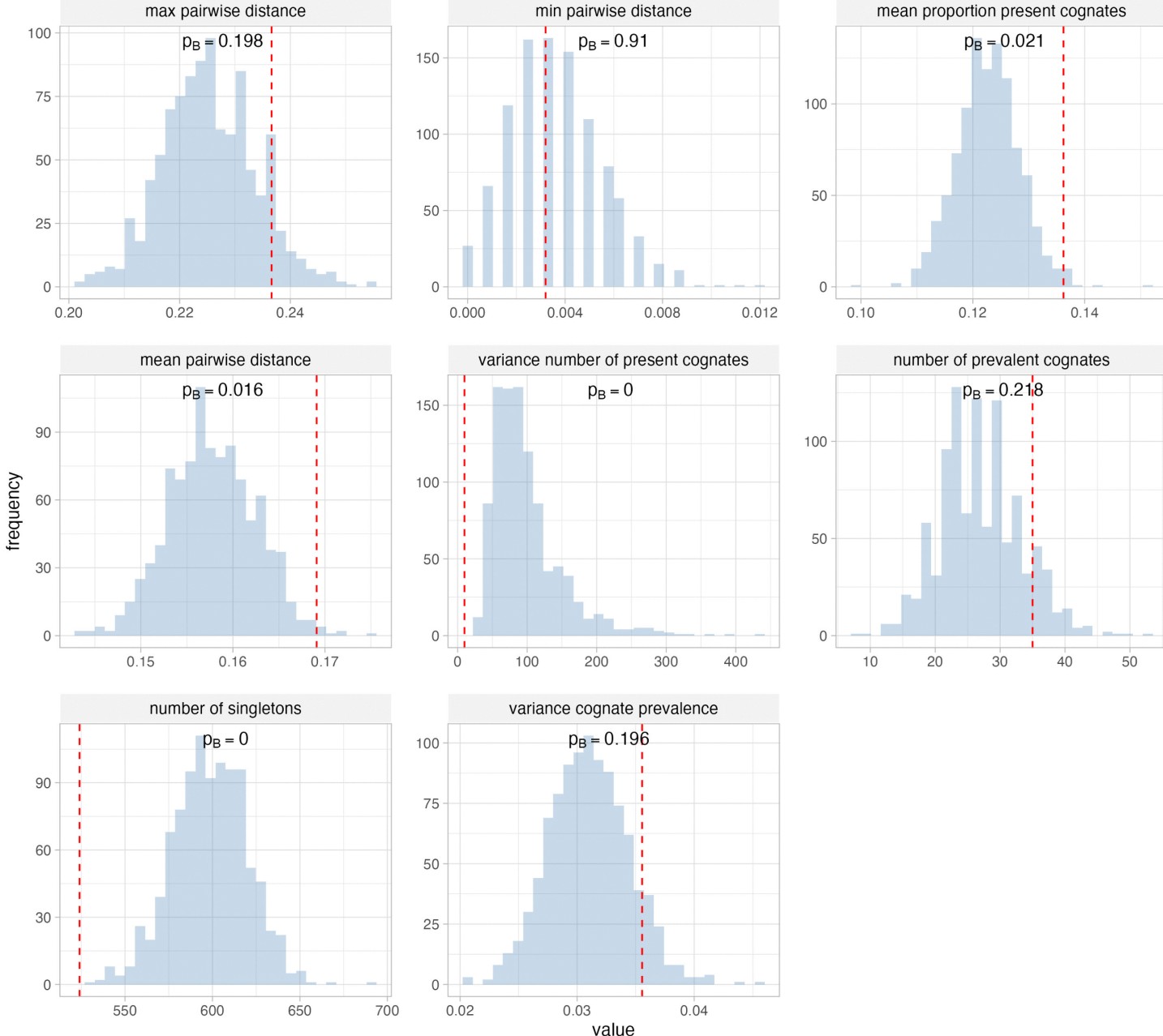

**Fig 6. Comparing eight metrics calculated on the posterior predictive datasets (blue histogram) and the empirical data (red line).** $p_B$ is the posterior predictive p value.

than that estimated by the analysis, and is in fact zero. When cognates are truly lost (rather than undergoing semantic shift) this process is not reversible. Over very long timescales any given cognate will eventually be lost, although many can persist for thousands of years.

The prevalence of semantic shift in the evolution of the lexicon could explain both the higher mean pairwise distance and the lower number of singletons in the empirical data. Frequent semantic shifts within a language subgroup could

inflate the number of cognate differences between closely related languages, but most of these "innovations" would be of cognate sets that occur elsewhere on the tree rather than new cognate sets unique to a single language.

Perhaps the least surprising result is the difference in the variance in the number of present cognates between the empirical and posterior predictive datasets. The IE-CoR dataset contains a single lexeme per meaning class per language, with few exceptions. Therefore cognate sets are not truly independent within a meaning class, despite being modelled as such. An alternative way of modelling the dataset is the multistate approach [3,19,34], where each meaning is modelled as a single character with multiple states, one for each cognate set. Most promising in this regard is the multistate model with infinite state space [34]. However, this has yet to be implemented on large-scale datasets or in mainstream phylogenetic software.

Model inadequacy, as shown by posterior predictive simulations, raises the question of how the estimation of the parameters of interest is affected. In a phylolinguistic analysis the parameters of interest are usually node ages and tree topology. Node age estimates could conceivably be affected by semantic shift, for example if cognate differences between closely related languages are inflated more relative to the differences between distantly languages. Node age estimates may also be affected by the non-independent evolution of cognates. This is demonstrated by previous implementations of a multistate model, which produced younger (arguably unrealistically young) node age estimates when compared to the binary covarion model [3,19,34]. Future work to explore how the inadequacy of the covarion model affects node age estimates is needed.

## Conclusion

The standard set-up for a phylogenetic analysis of linguistic data, combining ascertainment bias, partitioned rates and the covarion model has been validated. The BEAST2 software returns well-calibrated estimates of the covarion model substitution parameters and stationary frequencies. A caveat concerns the combination of ascertainment bias with partitions of varying length, a consequence of partition length being a direct consequence of the partition rate. Weighting the partition rates by the number of cognates, as is the default in BEAST2, results in the estimates of the partition rates and tree height/clock rate no longer being well-calibrated. Bayes Factors comparisons on empirical data confirm that assuming that each meaning comes has the same number of total cognate sets (including those that are absent in all languages), with partitions weighted by the number of meanings rather than the number of cognates, results in better fit. Weighting partitions by the number of meanings rather than the number of cognates should be the recommended practice in phylolinguistics going forward. An explicit weighted dirichlet distribution prior on the partition rates is also recommended, because an explicit prior distribution is lacking in the current BEAST2 default set-up.

Posterior predictive tests of model adequacy reveal misspecification of current phylogenetic models when it comes to capturing key aspects of lexical evolution. The prevalence of parallel semantic shift and the non-independent evolution of cognate sets are two major explanations for this. This highlights the need for more work on modelling lexical evolution in a realistic way. Researchers performing phylolinguistic analyses should be aware that treating each cognate set as independent, and the inability of current models to account for semantic shift, may distort results.

## Supporting information

**S1 Fig. Example of a PIT histogram, ECDF plot and ECDF difference plot for a well-calibrated model.**
(JPG)

**S2 Fig. Examples of ECDF difference plots for various examples of incorrectly implemented analyses.**
(JPG)

**S3 Fig. Coverage plot of validation simulation study 1.** Ascertainment cognates are not removed in this implementation.
(JPG)

**S4 Fig. ECDF difference plot of validation simulation study 1.** Ascertainment cognates are not removed in this implementation.
(JPG)

**S5 Fig. Parameter estimates under three weighting schemes for partition-specific rates.** From an analysis on Indo-Iranic languages.
(JPG)

**S6 Fig. Estimates of the partition-specific rate multipliers under three weighting schemes for partition-specific rates.** From an analysis on Indo-Iranic languages.
(JPG)

## Acknowledgments

I thank Alexei Drummond and Walter Xie for help with Lphy and members of the Department of Linguistic and Cultural Evolution for discussions.

## Author contributions

**Conceptualization:** Benedict King.

**Data curation:** Benedict King.

**Formal analysis:** Benedict King.

**Investigation:** Benedict King.

**Methodology:** Benedict King.

**Validation:** Benedict King.

**Visualization:** Benedict King.

**Writing – original draft:** Benedict King.

**Writing – review & editing:** Benedict King.

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
