## [Decision Letter · Decision Letter 0]

12 Mar 2026

PCOMPBIOL-D-25-02641

Testing the validity and adequacy of linguistic phylogenetic analyses

PLOS Computational Biology

Dear Dr. King,

Thank you for submitting your manuscript to PLOS Computational Biology. After careful consideration, we feel that it has merit but does not fully meet PLOS Computational Biology's publication criteria as it currently stands. Therefore, we invite you to submit a revised version of the manuscript that addresses the points raised during the review process.

We look forward to receiving your revised manuscript.

Kind regards,

Jordan Douglas

Academic Editor

PLOS Computational Biology

Zhaolei Zhang

Section Editor

PLOS Computational Biology

**Additional Academic Editor Comments:**

King has provided an important and much needed study on characterising the statistical properties of the widely-applied, but not well-understood, binary covarion model used in phylogenetic linguistics. The author has confirmed that the standard BEAST 2 implementation is indeed well-calibrated (using the standard methodologies used to validate most Bayesian phylogenetic models), but it is only valid under a certain set of conditions. The author also tested the adequacy of this method at explaining real data, finding that it does not capture some of the subtleties found in empirical datasets.

The paper is well-written and was met with overall positive feedback from three expert Reviewers. However, there are some major changes required before this manuscript can be recommended for publication.

Please note that there are 3 Reviewers, with the comments of Reviewer 3 uploaded separately as a PDF.

My own comments are below, coming from the perspective of a software developer for BEAST 2. In particular, I am concerned about the novelty behind some aspects of this work.

The reported issue behind partition weighting was already picked up by the core BEAST 2 team in mid 2025. The fix was released as part of BEAST 2.7.8 on June 18 2025. I am unsure what BEAST 2 version is being used in the present study. This issue behind Dirichlet priors and the weighted delta exchange operator has been around for a long time, and I believe it can even be traced back to before the BEAST/BEAST2 split. The relevant information on the fix is below:

BEAST 2.7.8 patch nodes:

https://www.beast2.org/2025/06/29/what-is-new-in-v2.7.8.html

GitHub issues:

https://github.com/BEAST2-Dev/BEASTLabs/issues/22

https://github.com/CompEvol/beast2/issues/1199

The experiments conducted here by the developers have not been published in any peer-reviewed journal setting, but there are simulation studies above similar to the ones conducted here by the author.

This aspect of the findings made in the present study therefore does not appear to be novel, in which case it would be a deduction against the overall level of contribution made by this study. The author should acknowledge these recent changes made to the BEAST 2 core, noting whether their own experiments were conducted before or after the fix, run new experiments where appropriate, and reevaluate the conclusions of the manuscript accordingly.

Minor issues

Line 44: I am not sure what beta refers to in this equation, and I cannot find any other uses of this variable in the manuscript.

Line 44: The author should also make it explicit here that Q describes the rate from the row to the column (not column to row as some fields prefer).

Lines 36-44: the author describes one commonly used parameterisation of the covarion model. But the BEAST 2 covarion implementation allows for three possible parameterisations, only one of which is described here. I suggest that all three parameterisations are acknowledged and contrasted, even if only one is being used in the rest of the paper. This will also be helpful for users who may be confused about this setting. The parameterisations are shown in the comments section at the top of the class below, and I have also copied them here for convenience:

https://github.com/CompEvol/beast2/blob/master/src/beast/base/evolution/substitutionmodel/BinaryCovarion.java

/**

* <p/>

* a    the rate of the slow rate class

* 1    the rate of the fast rate class

* p0    the equilibrium frequency of zero states

* p1    1 - p0, the equilibrium frequency of one states

* f0    the equilibrium frequency of slow rate class

* f1    1 - f0, the equilibrium frequency of fast rate class

* s, s1, s2 the rate of switching

* <p/>

* then the (unnormalized) instantaneous rate matrix (unnormalized Q) should be (depending on mode)

* <p/>

** mode = BEAST -- using classic BEAST implementation, reversible iff hfrequencies = (0.5, 0.5)

* FLAGS: reversible = false, TSParameterisation = false

** [ -(a*p1)-s ,   a*p1    ,    s   ,   0   ]

* [   a*p0    , -(a*p0)-s ,    0   ,   s   ]

* [    s      ,     0     ,  -p1-s ,  p1   ]

* [    0      ,     s     ,    p0  , -p0-s ]

** equilibrium frequencies

* [ p0 * f0, p1, * f0, p0 * f1, p1, * f1 ]

** mode = REVERSIBLE -- brings in hfrequencies in rate matrix

* reversible = true, TSParameterisation = false

* [ - , a , s , 0 ]

* [ a , - , 0 , s ]

* [ s , 0 , - , 1 ]

* [ 0 , s , 1 , - ]

** which with frequencies becomes

** [ -(a*p1*f0)-s*f0 ,   a*p1*f0       ,    s*f0      ,   0         ]

* [   a*p0*f0       , -(a*p0*f0)-s*f0 ,    0         ,   s*f0      ]

* [    s*f1         ,     0           ,  -p1*f1-s*f1 ,  p1*f1      ]

* [    0            ,     s*f1        ,    p0*f1     , -p0*f1-s*f1 ]

** equilibrium frequencies

* [ p0 * f0, p1, * f0, p0 * f1, p1, * f1 ]

** mode = TUFFLEYSTEEL uses alternative parameterisation: hfrequencies is ignored, and switch parameter is set to dimension = 2

* [ -(a*p1)-s1 ,   a*p1     ,    s1   ,   0    ]

* [   a*p0     , -(a*p0)-s1 ,    0    ,   s1   ]

* [    s2      ,     0      ,  -p1-s2 ,  p1    ]

* [    0       ,     s2     ,    p0   , -p0-s2 ]

** equilibrium frequencies

* [ p0 * s2/(s1+s2), p1, * s2/(s1+s2), p0 * s1/(s1+s2), p1, * s1/(s1+s2) ]

**

* Note: to use Tuffley & Steel's methods, set a = 0.

*/

Jordan Douglas

**Journal Requirements:**

**Reviewers' comments:**

Reviewer's Responses to Questions

**Comments to the Authors:**

Reviewer #1: This paper provides a much needed sanity check of the established “standard” methodology for Bayesian phylogenetics applied to linguistics, paying particular attention to the effects ascertainment bias on model inferences, as well as to the overall validity of the covarion model on linguistic data.

It provides a well motivated simulation study, seemingly one of the first studies specifically investigating the validity of models as they relate specifically to language/lexical evolution (though previous studies on the effects of lexical borrowing on tree topologies have been carried out e.g. Greenhill et al. 2009). The introduction presents a clear and fairly accessible summary of the key parts of the standard methodology which are later dissected in the paper, as well as a nice bibliography of phylolinguistic papers from the last 15 years or so.

The methods and discussion section is detailed, and comprehensive. A descriptive overview of the methods used to validate phylogenetic models is provided, and is easy to follow. It sets a standard for further model testing in phylolinguistics, even if the validation methodology has been presented elsewhere, and serves as a useful reference for computational historical linguists.

You have shown convincingly that the established model itself is identifiable so long as partitions are weighted by number of meanings not sites, and of course this is not difficult for future analyses to take into account. The discussion on the inadequacy of the binary covarion substitution model is interesting and articulates reasons why this may be the case. While the claim that the covarion is not a comprehensive or causally realistic model for “all aspects of lexical evolution” is likely uncontroversial, it's informative to have a more detailed explanation of the ways in which the model falls short. A short overview of current or future alternatives is included.

The code is all available and clear to read, as well as the BEAST2 .xmls for more practical instruction of how to implement the recommendations regarding partition weighting.

It is clear that bayesian phylogenetic analysis has become a indispensable tool for linguists of the modern era, and without sufficient validation of the methodology the field opens itself up to drawing conclusions based on model misspecification. This is particularly the case since the users of these methods often come from less technical backgrounds, and are more likely to use the default BEAST2 settings for example.

I enjoyed reading the paper, and have a few minor suggestions for improvement:

1. including a diagram of the covarion model could help to illustrate it in a more accessible way, though this is already done e.g. in the cited Hoffman et al. paper and elsewhere

2. In your SBC: simulating trees of 30 taxa according to a yule process seems reasonable to me, since you then test the adequacy according to a best-case scenario and show that under these ideal conditions, BEAST2's default partition weighting results in failed coverage and RUV checks. However, from a practical perspective, I would be interested to know how you expect the bias to scale with number of cognates and meanings, number of taxa, or tree calibrations.

3. I would appreciate a discussion of the impact of these findings on past research: particularly, how serious are the consequences of the ascertainment bias misspecification, are all trees with meaning rate heterogeneity/partitioning inferred in the literature of the past 10 years slightly longer than they should be etc.

4. The conclusion's point alluding to the covarion's “shortcomings” seem less severe than in both the abstract and in the previous discussion section “suggesting model misspecification”. I find myself wondering how severe it actually is, given its framing in the conclusion.

5. there are a few instances where “modelling” is spelt inconsistently, the sentence at lines 211-212 seems basically a duplicate of the one before, quotation marks are typographically wrong e.g. line 90 ”look like” instead of “look like”

Reviewer #2: This manuscript is a welcome and timely contribution to the methodology of phylogenetic inference as applied to linguistics, i.e. phylolinguistics. Its primary aim is to assess the validity and adequacy of phylogenetic models for languages using simulation-based tests. The study effectively demonstrates the limitations of current models and identifies key areas requiring further attention. The relevant literature is thoroughly and appropriately cited, and the methods employed are well-suited to address the research questions. As such, this work represents a valuable contribution to the field and should be published, provided the author addresses the points outlined below. Given that some revisions and additions may be necessary, I recommend "major revisions". However, I am confident that, once my suggestions are addressed, the manuscript will be ready for publication.

## Major suggestions

The title is too broad. It should explicitly state which specific aspects of validity and adequacy in phylolinguistic analyses are tested in the article.

If this article is aimed at linguists working on phylogenetic inference, it should be made more accessible. First, the distinction between validity and adequacy should be clarified with more details. What would it mean to have a valid but inadequate model, or conversely an invalid but adequate model? The author should give concrete recommandations or warnings about the consequences of the available methodological choices. The discussion and the conclusion are probably too short.

If this article is intended for linguists working on phylogenetic inference, it should be made more accessible. The distinction between validity and adequacy requires further clarification. What would it mean for a model to be valid but inadequate, or invalid but adequate? The author should provide concrete recommendations or warnings about the implications of methodological choices. The discussion and conclusion sections are too brief and should be expanded to address these points more thoroughly.

The rationale behind the Rank Uniformity Validation procedure should be explained in conceptual terms to aid understanding.

The terms "aligned" and "alignment" are misleading in a linguistic context. For example, on p. 5: "Validation simulation study 1, in which the simulated cognate alignments..." In phylolinguistics, "cognate alignment" typically refers to the alignment of sounds between word forms. While "alignment" and "sites" are standard terms in molecular phylogenetics, they do not directly apply to morphological or linguistic phylogenies. The author should rephrase these terms to avoid confusion.

Partitioning characters according to the number of cognate sets by meaning is common in phylolinguistic analyses. However, this raises the point of potential double dipping: we look at our particular data (cognate classes) to build a model and then use that model to analyse the same data. We can expect such a model to fit the data comparatively well, which is indeed the result quoted by the author. How much we should be concerned by this is beyond my expertise, but I know some colleagues are worried about this. It may be useful to discuss this point.

Partitioning characters based on the number of cognate sets by meaning is common in phylolinguistic analyses. However, this raises the issue of potential double-dipping: researchers use the same data (cognate classes) to both build a model and test it. Such models may naturally fit the data well, as results quoted by the author suggest. While the extent of this concern is beyond my expertise, I know that some colleagues are apprehensive about this practice. A discussion of this point would be valuable.

The author notes that empirical analyses rarely encounter all-absent cognate classes and typically include dummy columns in the character matrix to correct for missing data. However, the status of all-absent cognates in the simulation-based calibration study remains unclear. Were these generated because some cognate classes disappeared before reaching the tips during simulated evolution? Are these simply ascertainment bias correction columns? For Simulation Study 3, the partition rates were re-weighted, but the weighting criterion is not specified. It should also be made clear from the outset that this section focuses on testing the ascertainment bias correction procedure.

Figure 4 is difficult to interpret: What does the red staircase-like line represent? What does the purple cloud around the line signify?

Table 2 is also difficult to interpret: What is the denominator used to compute the Bayes factor? The large value in the Bayes factor column suggests that the "reweighted by number of cognates" model is the best, but this is not the case. Is there an error in the table?

## Minor comments

On p. 1, the author equates lexemes with word forms, but strictly speaking, a lexeme is an abstract unit defined as a set of word forms that share a lexical meaning and morphosyntactic properties, forming an inflectional paradigm. In practice, linguists record a representative form for the lexeme used to express a concept. A reference to literature describing the standard workflow (e.g., works by Greenhill et al. or List et al.) would be helpful.

On pp. 1–2: "Each cognate is treated as an independent character" should be clarified. More precisely, for binary-state characters, characters are pairs consisting of a concept and a cognate set.

On p. 2: "Cognates that are present in all languages, although invariant, can and are observed." While true, the author could add that this occurs because linguists typically use a standard list of concepts. In some language families, after data collection, researchers may observe no variation for a concept, i.e. a single cognate set exists. However, some studies choose to remove invariant characters, e.g. Goldstein, David. 2024. Divergence-time estimation in Indo-European. Diachronica 41(1). 1–45. https://doi.org/10.1075/dia.22031.gol.

On p. 5: "I extracted the subtree of these languages from the summary tree in [3], and fixed this during the analysis." It is unclear what was fixed. The author should specify this.

In Figures 2.jpg and S3 Fig.jpg, the blue and red colors are not visible in some panels because small intervals are obscured by black data points and the diagonal black line. To improve readability, the diagonal could be drawn with gray dots, and the data points could be represented as empty circles with outlines.

## Reproducibility

Instructions and comments for running the code are minimal (except for the simulation pipeline), and the project is difficult to navigate, making reproducibility challenging.

The provided R code relies on `setwd(dirname(rstudioapi::getSourceEditorContext()$path))`, which depends on RStudio and is not listed in the requirements. A project-oriented workflow (e.g., rstats.wtf/projects) would be preferable.

While the BEAST XML configuration files are provided, running all analyses is time-consuming. The author should also include the resulting log and tree files, as well as marginal_likelihood.txt and other output files. Currently, only the code to generate these files is provided. An undocumented and hidden file (IndoIranic_modeltesting/pathsampling/.RData) was mistakenly included and should be removed.

To ensure long-term reproducibility, the author should include a version of the asr_functions.R file in the project, rather than reading it from GitHub. This would allow the code to run offline, even if GitHub becomes unavailable or the file is modified or relocated.

**Have the authors made all data and (if applicable) computational code underlying the findings in their manuscript fully available?**

Reviewer #1: Yes

Reviewer #2: Yes

PLOS authors have the option to publish the peer review history of their article (what does this mean?). If published, this will include your full peer review and any attached files.

Reviewer #1: No

Reviewer #2: **Yes:** Thomas Pellard

**Figure resubmission:**
---

## [Decision Letter · Decision Letter 1]

10 May 2026

Dear Dr King,

We are pleased to inform you that your manuscript 'Testing the validity and adequacy of linguistic phylogenetic analyses' has been provisionally accepted for publication in PLOS Computational Biology.

Before your manuscript can be formally accepted you will need to complete some formatting changes, which you will receive in a follow up email. A member of our team will be in touch with a set of requests. **Specifically, please make sure to incorporate the minor corrections outlined below by the Academic Editor before publication.**

Best regards,

Jordan Douglas

Academic Editor

PLOS Computational Biology

Zhaolei Zhang

Section Editor

PLOS Computational Biology

**Comments from Academic Editor**

Thank you for your revised manuscript. It has been returned to the three original Reviewers, who are all satisfied with the changes. To avoid delays, I am recommending this manuscript for acceptance, but I request you make the following minor corrections prior to publication:

1. Please specify the BEAST version used in this study (2.7.8)

2. When you refer to the "default BEAST 2" settings, I assume you refer to any analysis that can be configured with beauti, as opposed to lphy or xml editing by hand. I suggest you make this explicit.

3. As picked up by Reviewer 1, make sure to correct lowercase "beast" to uppercase BEAST

4. Line 74: "The partition rates are drawn from Dirichlet distribution" -> missing article "a" before Dirichlet

Kind regards,

Jordan Douglas

Reviewer's Responses to Questions

**Comments to the Authors:**

Reviewer #1: I'm very happy with the changes made in the revision, many parts are clearer and I feel my points and those raised by the other reviewers have been adequately addressed.

In two places (line 57 and line 70) BEAST2 is written lowercase, but everywhere else is upper case.

Reviewer #2: The author has satisfactorily addressed all of my comments.

Reviewer #3: Congratulations on this nice piece of work. I'm happy to recommend acceptance.

**Have the authors made all data and (if applicable) computational code underlying the findings in their manuscript fully available?**

Reviewer #1: Yes

Reviewer #2: Yes

Reviewer #3: None

PLOS authors have the option to publish the peer review history of their article (what does this mean?). If published, this will include your full peer review and any attached files.

Reviewer #1: No

Reviewer #2: No

Reviewer #3: No

---

## [Editor Report · Acceptance letter]

PCOMPBIOL-D-25-02641R1

Testing the validity and adequacy of linguistic phylogenetic analyses

Dear Dr King,

I am pleased to inform you that your manuscript has been formally accepted for publication in PLOS Computational Biology. Your manuscript is now with our production department and you will be notified of the publication date in due course.

With kind regards,

Anita Estes
